# Charting brain growth and aging at high spatial precision

Saige Rutherford[1,2,3]*, Charlotte Fraza[1,2], Richard Dinga[1,2], Seyed Mostafa Kia[1,2,4], Thomas Wolfers[5,6], Mariam Zabihi[1,2], Pierre Berthet[5,6], Amanda Worker[7], Serena Verdi[8,9], Derek Andrews[10], Laura KM Han[11,12], Johanna MM Bayer[13,14], Paola Dazzan[7,15], Phillip McGuire[16], Roel T Mocking[17], Aart Schene[1,18], Chandra Sripada[3], Ivy F Tso[3], Elizabeth R Duval[3], Soo-Eun Chang[3], Brenda WJH Penninx[11,12], Mary M Heitzeg[3], S Alexandra Burt[19], Luke W Hyde[20], David Amaral[10], Christine Wu Nordahl[10], Ole A Andreasssen[6,21], Lars T Westlye[5,6,21], Roland Zahn[22], Henricus G Ruhe[1,18], Christian Beckmann[1,2,23†], Andre F Marquand[1,2†]

[1]Donders Institute for Brain, Cognition, and Behavior, Radboud University, Nijmegen, Netherlands; [2]Department of Cognitive Neuroscience, Radboud University Medical Center, Nijmegen, Netherlands; [3]Department of Psychiatry, University of Michigan, Ann Arbor, United States; [4]Department of Psychiatry, Utrecht University Medical Center, Utrecht, Netherlands; [5]Department of Psychology, University of Oslo, Oslo, Norway; [6]Norwegian Center for Mental Disorders Research (NORMENT), University of Oslo, and Oslo University Hospital, Oslo, Norway; [7]Department of Psychological Medicine, Institute of Psychiatry, Psychology and Neuroscience, King's College London, London, United Kingdom; [8]Centre for Medical Image Computing, Medical Physics and Biomedical Engineering, University College London, London, United Kingdom; [9]Dementia Research Centre, UCL Queen Square Institute of Neurology, London, United Kingdom; [10]The Medical Investigation of Neurodevelopmental Disorders (MIND) Institute and Department of Psychiatry and Behavioral Sciences, UC Davis School of Medicine, University of California, Davis, Sacramento, United States; [11]Amsterdam UMC, Vrije Universiteit, Psychiatry, Amsterdam Public Health Research Institute, Amsterdam, Netherlands; [12]GGZ inGeest, Amsterdam Neuroscience, Amsterdam, Netherlands; [13]Centre for Youth Mental Health, University of Melbourne, Melbourne, Australia; [14]Orygen Youth Health, Melbourne, Australia; [15]National Institute for Health Research Mental Health Biomedical Research Centre, South London and Maudsley National Health Service Foundation Trust and King's College London, London, United Kingdom; [16]Department of Psychosis Studies, Institute of Psychiatry, King's College London, London, United Kingdom; [17]Department of Psychiatry, Amsterdam UMC, Location AMC, Amsterdam, Netherlands; [18]Department of Psychiatry, Radboud University Medical Center, Nijmegen, Netherlands; [19]Department of Psychology, Michigan State University, East Lansing, United States; [20]Department of Psychology, University of Michigan, Ann Arbor, United States; [21]KG Jebsen Centre for Neurodevelopmental Disorders Research, Institute of Clinical Medicine, University of Oslo, Oslo, Norway; [22]Centre for Affective Disorders at the Institute of Psychiatry, King's College London, London, United Kingdom; [23]Centre for Functional MRI of the Brain (FMRIB), Nuffield Department of Clinical Neurosciences, Wellcome Centre for Integrative Neuroimaging, University of Oxford, Oxford, United Kingdom

*For correspondence:
saige.rutherford@donders.ru.nl

†These authors contributed equally to this work

**Abstract** Defining reference models for population variation, and the ability to study individual deviations is essential for understanding inter-individual variability and its relation to the onset and progression of medical conditions. In this work, we assembled a reference cohort of neuroimaging data from 82 sites (N=58,836; ages 2–100) and used normative modeling to characterize lifespan trajectories of cortical thickness and subcortical volume. Models are validated against a manually quality checked subset (N=24,354) and we provide an interface for transferring to new data sources. We showcase the clinical value by applying the models to a transdiagnostic psychiatric sample (N=1985), showing they can be used to quantify variability underlying multiple disorders whilst also refining case-control inferences. These models will be augmented with additional samples and imaging modalities as they become available. This provides a common reference platform to bind results from different studies and ultimately paves the way for personalized clinical decision-making.

## Editor's evaluation

This manuscript is of broad interest to the neuroimaging community. It establishes a detailed reference model of human brain development and lifespan trajectories based on a very large data set, across many cortical and subcortical brain regions. The model not only explains substantial variability on test data, it also successfully uncovers individual differences on a database of psychiatric patients that, in addition to group-level analyses, may be critical for diagnosis, thereby demonstrating high clinical potential. It presents a clear overview of the data resource, including detailed evaluation metrics, and makes code, models and documentation directly available to the community.

## Introduction

Since their introduction more than a century ago, normative growth charts have become fundamental tools in pediatric medicine and also in many other areas of anthropometry (*Cole, 2012*). They provide the ability to quantify individual variation against centiles of variation in a reference population, which shifts focus away from group-level (e.g., case-control) inferences to the level of the individual. This idea has been adopted and generalized in clinical neuroimaging, and normative modeling is now established as an effective technique for providing inferences at the level of the individual in neuroimaging studies (*Marquand et al., 2016*; *Marquand et al., 2019*).

Although normative modeling can be used to estimate many different kinds of mappings—for example between behavioral scores and neurobiological readouts—normative models of brain development and aging are appealing considering that many brain disorders are grounded in atypical trajectories of brain development (*Insel, 2014*) and the association between cognitive decline and brain tissue in aging and neurodegenerative diseases (*Jack et al., 2010*; *Karas et al., 2004*). Indeed, normative modeling has been applied in many different clinical contexts, including charting the development of infants born preterm (*Dimitrova et al., 2020*) and dissecting the biological heterogeneity across cohorts of individuals

**Table 1.** Sample description and demographics.
mQC refers to the manual quality checked subset of the full sample. 'All' rows=Train+Test. Clinical refers to the transdiagnostic psychiatric sample (diagnostic details in *Figure 2A*).

|         |              | N (subjects) | N (sites) | Sex (%F/%M) | Age (Mean, **S.D**) |
|---------|--------------|--------------|-----------|-------------|----------------------|
| Full    | All          | 58,836       | 82        |             |                      |
|         | Training set | 29,418       | 82        | 51.1/48.9   | 46.9, 24.4           |
|         | Test set     | 29,418       | 82        | 50.9/49.1   | 46.9, 24.4           |
| mQC     | All          | 24,354       | 59        |             |                      |
|         | Training set | 12,177       | 59        | 50.2/49.8   | 30.2, 24.1           |
|         | Test set     | 12,177       | 59        | 50.4/49.4   | 30.1, 24.2           |
| Clinical | Test set    | 1985         | 24        | 38.9/61.1   | 30.5, 14.1           |
| Transfer | Test set    | 546          | 6         | 44.5/55.5   | 24.8, 13.7           |

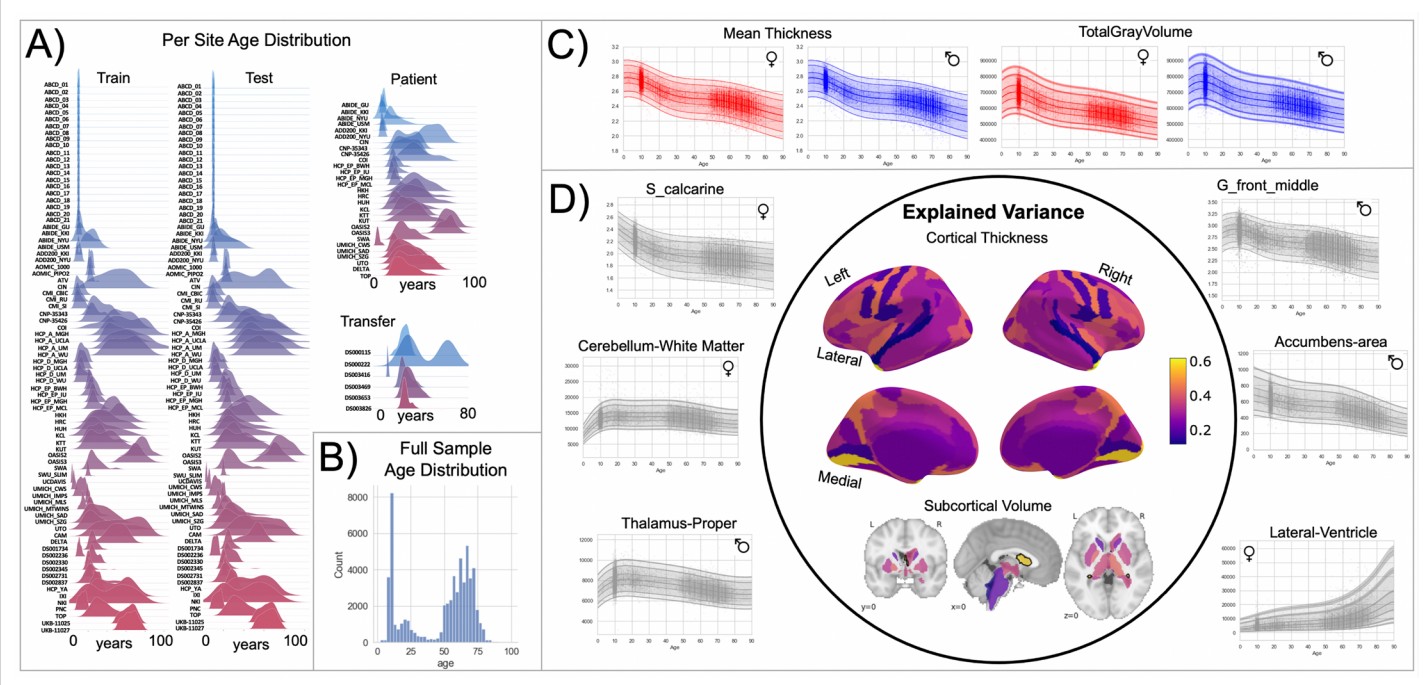

**Figure 1.** Normative model overview. (**A**) Age density distribution (x-axis) of each site (y-axis) in the full model train and test, clinical, and transfer validation set. (**B**) Age count distribution of the full sample (N=58,836). (**C, D**) Examples of lifespan trajectories of brain regions. Age is shown on x-axis and predicted thickness (or volume) values are on the y-axis. Centiles of variation are plotted for each region. In (**C**), we show that sex differences between females (red) and males (blue) are most pronounced when modeling large-scale features such as mean cortical thickness across the entire cortex or total gray matter volume. These sex differences manifest as a shift in the mean in that the shape of these trajectories is the same for both sexes, as determined by sensitivity analyses where separate normative models were estimated for each sex. The explained variance (in the full test set) of the whole cortex and subcortex is highlighted inside the circle of (**D**). All plots within the circle share the same color scale. Visualizations for all ROI trajectories modeled are shared on GitHub for users that wish to explore regions not shown in this figure.

with different brain disorders, including schizophrenia, bipolar disorder, autism, and attention-deficit/hyperactivity disorder (*Bethlehem et al., 2020*; *Wolfers et al., 2021*; *Zabihi et al., 2019*).

A hurdle to the widespread application of normative modeling is a lack of well-defined reference models to quantify variability across the lifespan and to compare results from different studies. Such models should: (1) accurately model population variation across large samples; (2) be derived from widely accessible measures; (3) provide the ability to be updated as additional data come online, (4) be supported by easy-to-use software tools, and (5) should quantify brain development and aging at a high spatial resolution, so that different patterns of atypicality can be used to stratify cohorts and predict clinical outcomes with maximum spatial precision. Prior work on building normative modeling reference cohorts (*Bethlehem et al., 2021*) has achieved some of these aims (1–4), but has modeled only global features (i.e., total brain volume), which is useful for quantifying brain growth but has limited utility for the purpose of stratifying clinical cohorts (aim 5). The purpose of this paper is to introduce a set of reference models that satisfy all these criteria.

To this end, we assemble a large neuroimaging data set (*Table 1*) from 58,836 individuals across 82 scan sites covering the human lifespan (aged 2–100, *Figure 1A*) and fit normative models for cortical thickness and subcortical volumes derived from Freesurfer (version 6.0). We show the clinical utility of these models in a large transdiagnostic psychiatric sample (N=1985, *Figure 2*). To maximize the utility of this contribution, we distribute model coefficients freely along with a set of software tools to enable researchers to derive subject-level predictions for new data sets against a set of common reference models.

## Results

We split the available data into training and test sets, stratifying by site (*Table 1*, *Supplementary files 1 and 2*). After careful quality checking procedures, we fit normative models using a set of covariates

## A) Clinical Validation Sample Description

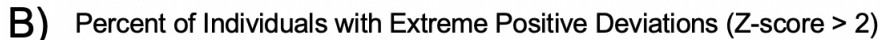

|  | N (subjects) | N (sites) | Sex (%F/%M) | Age (m, s.d) |
|---|---|---|---|---|
| Attention Deficit Disorder (ADHD) | 111 | 4 | 36/64 | 17.6, 12.4 |
| Autism Spectrum Disorder (ASD) | 450 | 6 | 14/86 | 19.1, 11.1 |
| Early Psychosis (EP) | 207 | 5 | 39/61 | 25.6, 7.2 |
| Schizophrenia (SZ) | 469 | 8 | 41/59 | 31.9, 9.5 |
| Bipolar Disorder (BD) | 249 | 4 | 55/45 | 34.5, 10.9 |
| Major Depression Disorder (MDD) | 499 | 9 | 52/48 | 40.2, 13.4 |

## B) Percent of Individuals with Extreme Positive Deviations (Z-score > 2)

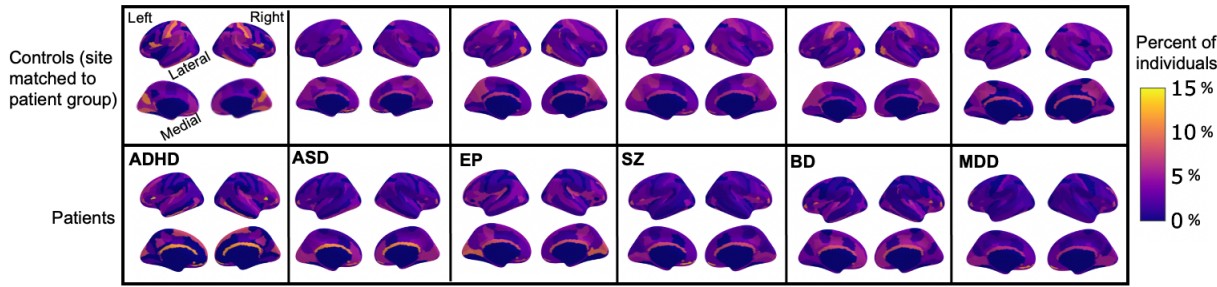

## C) Percent of Individuals with Extreme Negative Deviations (Z-score < -2)

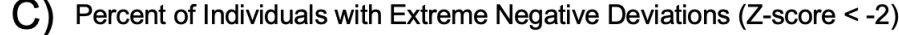

## D) Control vs. Patient Univariate T-Tests

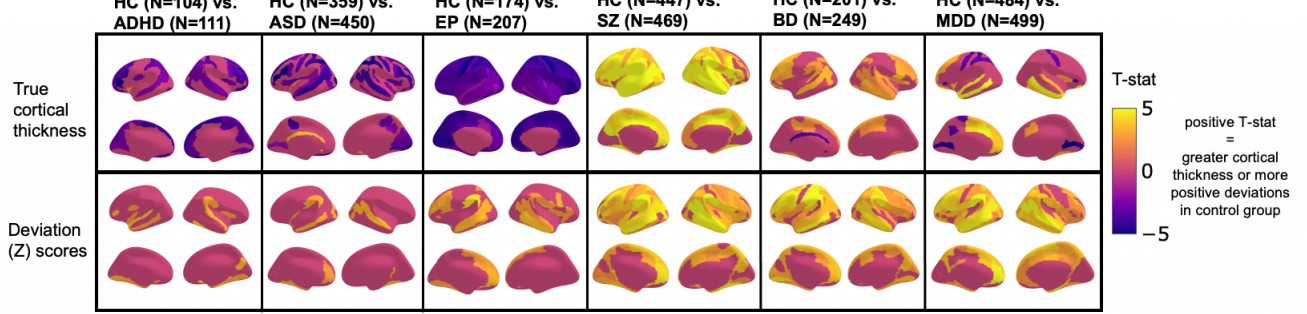

**Figure 2.** Normative modeling in clinical cohorts. Reference brain charts were transferred to several clinical samples (described in (**A**)). Patterns of extreme deviations were summarized for each clinical group and compared to matched control groups (from the same sites). (**B**) Shows extreme positive deviations (thicker/larger than expected) and (**C**) shows the extreme negative deviation (thinner/smaller than expected) patterns. (**D**) Shows the significant (FDR corrected p<0.05) results of classical case-control methods (mass-univariate t-tests) on the true cortical thickness data (top row) and on the deviations scores (bottom row). There is unique information added by each approach which becomes evident when noticing the maps in (**B–D**) are not identical. ADHD, attention-deficit hyperactive disorder; ASD, autism spectrum disorder; BD, bipolar disorder; EP, early psychosis; FDR, false discovery rate; MDD, major depressive disorder; SZ, schizophrenia.

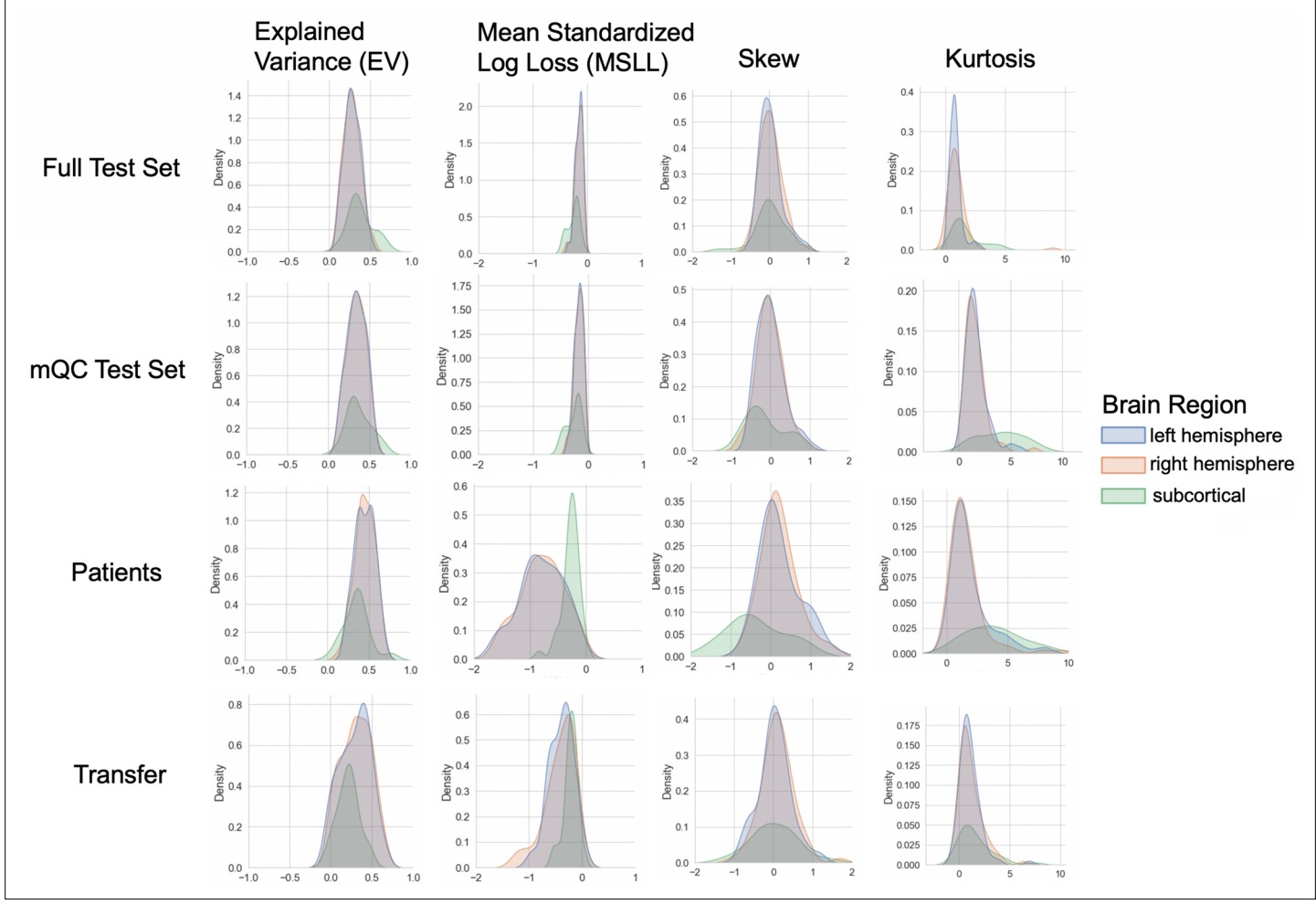

**Figure 3.** Evaluation metrics across all test sets. The distribution of evaluation metrics in four different test sets (full, mQC, patients, and transfer, see Materials and methods) separated into left and right hemispheres and subcortical regions, with the skew and excess kurtosis being measures that depict the accuracy of the estimated shape of the model, ideally both would be around zero. Note that kurtosis is highly sensitive to outlying samples. Overall, these models show that the models fit well in term of central tendency and variance (explained variance and MSLL) and model the shape of the distribution well in most regions (skew and kurtosis). Code and sample data for transferring these models to new sites not included in training is shared.

The online version of this article includes the following figure supplement(s) for figure 3:

**Figure supplement 1.** Comparison of the explained variance in cortical thickness across the different test sets.

**Figure supplement 2.** Showing the explained variance for each brain region across 10 randomized resampling of the full control test set.

**Figure supplement 3.** Per site explained variance across the different test sets.

(age, sex, and fixed effects for site) to predict cortical thickness and subcortical volume for each parcel in a high-resolution atlas (*Destrieux et al., 2010*). We employed a warped Bayesian linear regression model to accurately model non-linear and non-Gaussian effects (*Fraza et al., 2021*), whilst accounting for scanner effects (*Bayer et al., 2021*; *Kia et al., 2021*). These models are summarized in *Figure 1* and *Figure 3*, *Figure 3—figure supplements 1–3*, and with an online interactive visualization tool for exploring the evaluation metrics across different test sets (overview of this tool shown in *Video 1*). The raw data used in these visualizations are available on GitHub (*Rutherford, 2022a*).

We validate our models with several careful procedures: first, we report out of sample metrics; second, we perform a supplementary analysis on a subset of participants for whom input data had undergone manual quality checking by an expert rater (*Table 1* – mQC). Third, each model fit was evaluated using metrics (*Figure 3*, *Figure 3—figure supplements 1–3*) that quantify central tendency and distributional accuracy (*Dinga et al., 2021*; *Fraza et al., 2021*). We also estimated separate models for males and females, which indicate that sex effects are adequately modeled using a global offset.

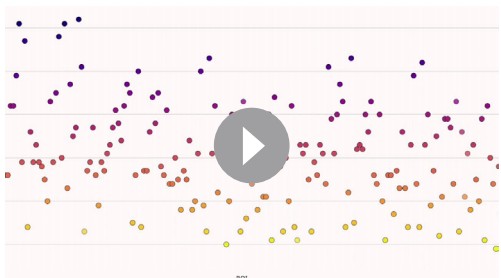

**Video 1.** "Demonstration of the functionality of our interactive online visualization tool (https://brainviz-app.herokuapp.com/) that is available for all evaluation metrics across all test sets. The code for creating this website can be found on GitHub (https://github.com/saigerutherford/brainviz-app; copy archived at swh:1:rev:021fff9a48b26f2d07bbb4b3fb92cd5202418905; *Rutherford, 2022b*).
https://elifesciences.org/articles/72904/figures#video1

Finally, to facilitate independent validation, we packaged pretrained models and code for transferring to new samples into an open resource for use by the community and demonstrated how to transfer the models to new samples (i.e., data not present in the initial training set).

Our models provide the opportunity for mapping the diverse trajectories of different brain areas. Several examples are shown in *Figure 1C and D* which align with known patterns of development and aging (*Ducharme et al., 2016*; *Gogtay et al., 2004*; *Tamnes et al., 2010*). Moreover, across the cortex and subcortex our model fits well, explaining up to 80% of the variance out of sample (*Figure 3*, *Figure 3—figure supplements 1–3*).

A goal of this work is to develop normative models that can be applied to many different clinical conditions. To showcase this, we apply the model to a transdiagnostic psychiatric cohort (*Table 1* – Clinical; *Figure 2*) resulting in personalized, whole-brain deviation maps that can be used to understand inter-individual variability (e.g., for stratification) and to quantify group separation (e.g., case-control effects). To demonstrate this, for each clinical group, we summarized the individual deviations within that group by computing the proportion of subjects that have deviations in each region and comparing to matched (same sites) controls in the test set (*Figure 2B–C*). Additionally, we performed case-control comparisons on the raw cortical thickness and subcortical volumes, and on the deviation maps (*Figure 2D*), again against a matched sample from the test set. This demonstrates the advantages of using normative models for investigating individual differences in psychiatry, that is, quantifying clinically relevant information at the level of each individual. For most diagnostic groups, the z-statistics derived from the normative deviations also provided stronger case-control effects than the raw data. This shows the importance of accurate modeling of population variance across multiple clinically relevant dimensions. The individual-level deviations provide complimentary information to the group effects, which aligns with previous work (*Wolfers et al., 2018*; *Wolfers et al., 2020*; *Zabihi et al., 2020*). We note that a detailed description of the clinical significance of our findings is beyond the scope of this work and will be presented separately.

## Discussion

In this work, we create lifespan brain charts of cortical thickness and subcortical volume derived from structural MRI, to serve as reference models. Multiple data sets were joined to build a mega-site lifespan reference cohort to provide good coverage of the lifespan. We applied the reference cohort models to clinical data sets and demonstrated the benefits of normative modeling in addition to standard case-control comparisons. All models, including documentation and code, are made available to the research community. We also provide an example data set (that includes data from sites not in the training sample) along with the code to demonstrate how well our models can adapt to new sites, and how easy it is to transfer our pretrained models to users' own data sets.

We identify three main strengths of our approach. First, our large lifespan data set provides high anatomical specificity, necessary for discriminating between conditions, predicting outcomes, and stratifying subtypes. Second, our models are flexible in that they can model non-Gaussian distributions, can easily be transferred to new sites, and are built on validated analytical techniques and software tools (*Fraza et al., 2021*; *Kia et al., 2021*; *Marquand et al., 2019*). Third, we show the general utility of this work in that it provides the ability to map individual variation whilst also improving case-control inferences across multiple disorders.

In recent work, a large consortium established lifespan brain charts that are complementary to our approach (*Bethlehem et al., 2021*). Benefits of their work include precisely quantifying brain growth using a large cohort, but they only provide estimates of four coarse global measures (e.g., total brain

volume). While this can precisely quantify brain growth and aging this does not provide the ability to generate individualized fingerprints or to stratify clinical cohorts. In contrast, in this work, we focus on providing spatially specific estimates (188 different brain regions) across the post-natal lifespan which provides fine-grained anatomical estimates of deviation, offering an individualized perspective that can be used for clinical stratification. We demonstrate the transdiagnostic clinical value of our models (*Figure 2*) by showing how clinical variation is widespread in a fine-grain manner (e.g., not all individuals deviate in the same regions and not all disorders have the same characteristic patterns) and we facilitate clinical applications of our models by sharing tutorial code notebooks with sample data that can be run locally or online in a web browser.

We also identify the limitations of this work. We view the word 'normative' as problematic. This language implies that there are normal and abnormal brains, a potentially problematic assumption. As indicated in *Figure 2*, there is considerable individual variability and heterogeneity among trajectories. We encourage the use of the phrase 'reference cohort' over 'normative model'. In order to provide coverage of the lifespan the curated data set is based on aggregating existing data, meaning there is unavoidable sampling bias. Race, education, and socioeconomic variables were not fully available for all included data sets, however, given that data were compiled from research studies, they are likely samples drawn predominantly from Western, Educated, Industrialized, Rich, and Democratic (WEIRD) societies (*Henrich et al., 2010*) and future work should account for these factors. The sampling bias of UKBiobank (*Fry et al., 2017*) is especially important for users to consider as UKBiobank data contributes 59% of the full sample. By sampling both healthy population samples and case-control studies, we achieve a reasonable estimate of variation across individuals, however, downstream analyses should consider the nature of the reference cohort and whether it is appropriate for the target sample. Second, we have relied on semi-automated quality control (QC) for the full sample which—despite a conservative choice of inclusion threshold—does not guarantee either that low-quality data were excluded or that the data were excluded are definitively excluded because of artifacts. We addressed this by comparing our full test set to a manually quality check data set and observed similar model performance. Also, Freesurfer was not adjusted for the very young age ranges (2–7 yo) thus caution should be used when interpreting the model on new data in this age range. Finally, although the models presented in this study are comprehensive, they are only the first step, and we will augment our repository with more diverse data, different features, and modeling advances as these become available.

## Materials and methods

Data from 82 sites were combined to create the initial full sample. These sites are described in detail in *Supplementary files 1-2*, including the sample size, age (mean and standard deviation), and sex distribution of each site. Many sites were pulled from publicly available data sets including ABCD, ABIDE, ADHD200, CAMCAN, CMI-HBN, HCP-Aging, HCP-Development, HCP-Early Psychosis, HCP-Young Adult, IXI, NKI-RS, Oasis, OpenNeuro, PNC, SRPBS, and UKBiobank. For data sets that include repeated visits (i.e., ABCD and UKBiobank), only the first visit was included. Other included data come from studies conducted at the University of Michigan (*Duval et al., 2018*; *Rutherford et al., 2020*; *Tomlinson et al., 2020*; *Tso et al., 2021*; *Weigard et al., 2021*; *Zucker et al., 2009*), University of California Davis (*Nordahl et al., 2020*), University of Oslo (*Nesvåg et al., 2017*), King's College London (*Green et al., 2012*; *Lythe et al., 2015*), and Amsterdam University Medical Center (*Mocking et al., 2016*). Full details regarding sample characteristics, diagnostic procedures, and acquisition protocols can be found in the publications associated with each of the studies. Equal sized training and testing data sets (split half) were created using scikit-learn's train_test_split function, stratifying on the site variable. It is important to stratify based on site, not only study (*Bethlehem et al., 2021*), as many of the public studies (i.e., ABCD) include several sites, thus modeling study does not adequately address MRI scanner confounds. To test stability of the model performance, the full test set was randomly resampled 10 times and evaluation metrics were re-calculated on each split of the full test set (*Figure 3—figure supplement 2*). To show generalizability of the models to new data not included in training, we leveraged data from OpenNeuro.org (*Markiewicz et al., 2021*) to create a transfer data set (six sites, N=546, *Supplementary file 3*). This data are provided along with the code for transferring to walk users through how to apply these models to their own data.

The clinical validation sample consisted of a subset of the full data set (described in detail in *Figure 1A*, *Figure 2A* and *Supplementary file 1*). Studies (sites) contributing clinical data included: Autism Brain Imaging Database Exchange (ABIDE GU, KKI, NYU, USM), ADHD200 (KKI, NYU), CNP, SRPBS (CIN, COI, KTT, KUT, HKH, HRC, HUH, SWA, UTO), Delta (AmsterdamUMC), Human Connectome Project Early Psychosis (HCP-EP BWH, IU, McL, MGH), KCL, University of Michigan Children Who Stutter (UMich_CWS), University of Michigan Social Anxiety Disorder (UMich_SAD), University of Michigan Schizophrenia Gaze Processing (UMich_SZG), and TOP (University of Oslo).

In addition to the sample-specific inclusion criteria, inclusion criteria for the full sample were based on participants having basic demographic information (age and sex), a T1-weighted MRI volume, and Freesurfer output directories that include summary files that represent left and right hemisphere cortical thickness values of the Destrieux parcellation and subcortical volumetric values (aseg.stats, lh.aparc.a2009s.stats, and rh.aparc.a2009s.stats). Freesurfer image analysis suite (version 6.0) was used for cortical reconstruction and volumetric segmentation for all studies. The technical details of these procedures are described in prior publications (*Dale et al., 1999*; *Fischl et al., 2002*; *Fischl and Dale, 2000*). UK Biobank was the only study for which Freesurfer was not run by the authors. Freesurfer functions *aparcstats2table* and *asegstats2table* were run to extract cortical thickness from the Destrieux parcellation (*Destrieux et al., 2010*) and subcortical volume for all participants into CSV files. These files were inner merged with the demographic files, using Pandas, and NaN rows were dropped.

QC is an important consideration for large samples and is an active research area (*Alfaro-Almagro et al., 2018*; *Klapwijk et al., 2019*; *Rosen et al., 2018*). We consider manual quality checking of images both prior to and after preprocessing to be the gold standard. However, this is labor intensive and prohibitive for very large samples. Therefore, in this work, we adopt a pragmatic and multi-pronged approach to QC. First, a subset of the full data set underwent manual quality checking (mQC) by author S.R. Papaya, a JavaScript-based image viewer. Manual quality checking was performed during December 2020 when the Netherlands was in full lockdown due to COVID-19 and S.R. was living alone in a new country with a lot of free time. Data included in this manual QC step was based on what was available at the time (*Supplementary file 2*). Later data sets that were included were not manually QC'd due to resource and time constraints. Scripts were used to initialize a manual QC session and track progress and organize ratings. All images (T1w volume and Freesurfer brain.finalsurfs) were put into JSON files that the mQC script would call when loading Papaya. Images were rated using a 'pass/fail/flag' scale and the rating was tracked in an automated manner using keyboard inputs (up arrow=pass, down arrow=fail, F key=flag, and left/right arrows were used to move through subjects). Each subject's T1w volume was viewed in 3D volumetric space, with the Freesurfer brain.finalsurfs file as an overlay, to check for obvious quality issues such as excessive motion, ghosting or ringing artifacts. Example scripts used for quality checking and further instructions for using the manual QC environment can be found on GitHub(*Rutherford, 2022c* copy archived at swh:1:rev:70894691c74febe2a4d40ab0c84c50094b9e99ce). We relied on ABCD consortium QC procedures for the QC for this sample. The ABCD study data distributes a variable (freesqc01.txt; fsqc_qc = = 1/0) that represents manual quality checking (pass/fail) of the T1w volume and Freesurfer data, thus this data set was added into our manual quality checked data set bringing the sample size to 24,354 individuals passing manual quality checks. Note that QC was performed on the data prior to splitting of the data to assess generalizability. Although this has a reduced sample, we consider this to be a gold-standard sample in that every single scan has been checked manually. All inferences reported in this manuscript were validated against this sample. Second, for the full sample, we adopted an automated QC procedure that quantifies image quality based on the Freesurfer Euler Characteristic (EC), which has been shown to be an excellent proxy for manual labeling of scan quality (*Monereo-Sánchez et al., 2021*; *Rosen et al., 2018*) and is the most important feature in automated scan quality classifiers (*Klapwijk et al., 2019*). Since the distribution of the EC varies across sites, we adopt a simple approach that involves scaling and centering the distribution over the EC across sites and removing samples in the tail of the distribution (see *Kia et al., 2021* for details). While any automated QC heuristic is by definition imperfect, we note that this is based on a conservative inclusion threshold such that only samples well into the tail of the EC distribution are excluded, which are likely to be caused by true topological defects rather than abnormalities due to any underlying pathology. We separated the evaluation metrics into full test set (relying on automated QC) and mQC test set in order to compare model performance between

the two QC approaches and were pleased to notice that the evaluation metrics were nearly identical across the two methods.

Normative modeling was run using python 3.8 and the PCNtoolkit package (version 0.20). Bayesian Linear Regression (BLR) with likelihood warping was used to predict cortical thickness and subcortical volume from a vector of covariates (age, sex, and site). For a complete mathematical description and explanation of this implementation, see *Fraza et al., 2021*. Briefly, for each brain region of interest (cortical thickness or subcortical volume), $y$ is predicted as:

$$y = w^T \phi\left(x\right) + \epsilon \tag{1}$$

where $w^T$ is the estimated weight vector, $\phi\left(x\right)$ is a basis expansion of the of covariate vector **x,** consisting of a B-spline basis expansion (cubic spline with five evenly spaced knots) to model non-linear effects of age, and $\epsilon = \eta\left(0, \beta\right)$ a Gaussian noise distribution with mean zero and noise precision term β (the inverse variance). A likelihood warping approach (*Rios and Tobar, 2019*; *Snelson et al., 2003*) was used to model non-Gaussian effects. This involves applying a bijective non-linear warping function to the non-Gaussian response variables to map them to a Gaussian latent space where inference can be performed in closed form. We employed a 'sinarcsinsh' warping function, which is equivalent to the SHASH distribution commonly used in the generalized additive modeling literature (*Jones and Pewsey, 2009*) and which we have found to perform well in prior work (*Dinga et al., 2021*; *Fraza et al., 2021*). Site variation was modeled using fixed effects, which we have shown in prior work provides relatively good performance (*Kia et al., 2021*), although random effects for site may provide additional flexibility at higher computational cost. A fast numerical optimization algorithm was used to optimize hyperparameters (L-BFGS). Computational complexity of hyperparameter optimization was controlled by minimizing the negative log-likelihood. Deviation scores (Z-scores) are calculated for the n-th subject, and d-th brain area, in the test set as:

$$Z_{nd} = \frac{y_{nd} - \hat{y}_{nd}}{\sqrt{\sigma_d^2 + (\sigma_*^2)_d}} \tag{2}$$

Where $y_{nd}$ is the true response, $\hat{y}_{nd}$ is the predicted mean, $\sigma_d^2$ is the estimated noise variance (reflecting uncertainty in the data), and $\left(\sigma^2\right)_d$ is the variance attributed to modeling uncertainty. Model fit for each brain region was evaluated by calculating the explained variance (which measures central tendency), the mean squared log-loss (MSLL, central tendency, and variance) plus skew and kurtosis of the deviation scores (2) which measures how well the shape of the regression function matches the data (*Dinga et al., 2021*). Note that for all models, we report out of sample metrics.

To provide a summary of individual variation within each clinical group, deviation scores were summarized for each clinical group (*Figure 2B–C*) by first separating them into positive and negative deviations, counting how many subjects had an extreme deviation (positive extreme deviation defined as Z>2, negative extreme deviation as Z<−2) at a given ROI, and then dividing by the group size to show the percentage of individuals with extreme deviations at that brain area. Controls from the same sites as the patient groups were summarized in the same manner for comparison. We also performed classical case versus control group difference testing on the true data and on the deviation scores (*Figure 2D*) and thresholded results at a Benjamini-Hochberg false discovery rate of p<0.05. Note that in both cases, we directly contrast each patient group to their matched controls to avoid nuisance variation confounding any reported effects (e.g., sampling characteristics and demographic differences).

All pretrained models and code are shared online with straightforward directions for transferring to new sites and including an example transfer data set derived from several OpenNeuro.org data sets. Given a new set of data (e.g., sites not present in the training set), this is done by first applying the warp parameters estimating on the training data to the new data set, adjusting the mean and variance in the latent Gaussian space, then (if necessary) warping the adjusted data back to the original space, which is similar to the approach outlined in *Dinga et al., 2021*. Note that to remain unbiased, this should be done on a held-out calibration data set. To illustrate this procedure, we apply this approach to predicting a subset of data that was not used during the model estimation step. We leveraged data from OpenNeuro.org (*Markiewicz et al., 2021*) to create a transfer data set (six sites, N=546, *Supplementary file 3*). This data are provided along with the code for transferring to walk users through how to apply these models to their own data. These results are reported in *Figure 3* (transfer) and *Supplementary file 3*. We also distribute scripts for this purpose in the GitHub Repository associated

with this manuscript. Furthermore, to promote the use of these models and remove barriers to using them, we have set up access to the pretrained models and code for transferring to users' own data, using Google Colab, a free, cloud-based platform for running python notebooks. This eliminates the need to install python/manage package versions and only requires users to have a personal computer with stable internet connection.

## Acknowledgements

This research was supported by grants from the European Research Council (ERC, grant 'MENTALPRE-CISION' 10100118 and 'BRAINMINT' 802998), the Wellcome Trust under an Innovator award ('BRAIN-CHART,' 215698/Z/19/Z) and a Strategic Award (098369/Z/12/Z), the Dutch Organisation for Scientific Research (VIDI grant 016.156.415) the Research Council of Norway (223273, 249795, 298646, 300768, and 276082), the South-Eastern Norway Regional Health Authority (2014097, 2015073, 2016083, and 2019101), the KG Jebsen Stiftelsen, an Autism Center of Excellence grant awarded by the National Institute of Child Health and Development (NICHD) (P50 HD093079) as well as the National Institute of Mental Health (R01MH104438 and R01MH103371). TW also gratefully acknowledges the Niels Stensen Fellowship as well as the European Union's Horizon 2020 Research and Innovation Program under the Marie Sklodowska-Curie Grant Agreement no. 895011. RZ was funded by Medical Research Council grant (G0902304). IFT was funded by National Institute of Mental Health K23MH108823. SC was funded by National Institute on Deafness and other Communication Disorders (NIDCD/NIH) grant R01DC011277. CS was funded by the National Institute of Mental Health R01MH107741. LD was funded by Michigan Institute for Clinical Health Research (MICHR) Pilot Grant Program (UL1TR002240) through an NIH Clinical and Translational Science Award (CTSA). MTwiNS was supported by the National Institute of Mental Health and the Office of the Director National Institute of Health, under Award Number UG3MH114249 and the Eunice Kennedy Shriver National Institute of Child Health & Human Development of the National Institutes of Health under Award number R01HD093334 to SAB and LWH. RJTM was funded by an ABC Talent Grant. The ABCD Study is supported by the National Institutes of Health (NIH) and additional federal partners under Award numbers U01DA041022, U01DA041028, U01DA041048, U01DA041089, U01DA041106, U01DA041117, U01DA041120, U01DA041134, U01DA041148, U01DA041156, U01DA041174, U24DA041123, and U24DA041147.

## Additional information

### Competing interests

Ole A Andreasssen: is a consultant for HealthLytix and received speaker's honorarium from Lundbeck and Sunovion. Henricus G Ruhe: received speaker's honorarium from Lundbeck and Janssen. Christian Beckmann: is director and shareholder of SBGNeuro Ltd. The other authors declare that no competing interests exist.

### Funding

| Funder | Grant reference number | Author |
| --- | --- | --- |
| H2020 European Research Council | 10100118 | Andre F Marquand |
| H2020 European Research Council | 802998 | Lars T Westlye |
| Wellcome Trust | 215698/Z/19/Z | Andre F Marquand |
| Wellcome Trust | 098369/Z/12/Z | Christian Beckmann |
| Nederlandse Organisatie voor Wetenschappelijk Onderzoek | VIDI grant 016.156.415 | Andre F Marquand |
| National Institute of Mental Health | R01MH104438 | David Amaral Christine Wu Nordahl |

| Funder | Grant reference number | Author |
|---|---|---|
| National Institute of Mental Health | R01MH103371 | David Amaral Christine Wu Nordahl |
| Eunice Kennedy Shriver National Institute of Child Health and Human Development | P50 HD093079 | David Amaral Christine Wu Nordahl |
| H2020 Marie Skłodowska-Curie Actions | 895011 | Thomas Wolfers |
| Medical Research Council | G0902304 | Roland Zahn |
| National Institute of Mental Health | K23MH108823 | Ivy F Tso |
| National Institute on Deafness and Other Communication Disorders | R01DC011277 | Soo-Eun Chang |
| National Institute of Mental Health | R01MH107741 | Chandra Sripada |
| Michigan Institute for Clinical and Health Research | UL1TR002240 | Elizabeth R Duval |
| National Institute of Mental Health | UG3MH114249 | S Alexandra Burt Luke Hyde |
| Eunice Kennedy Shriver National Institute of Child Health and Human Development | R01HD093334 | S Alexandra Burt Luke Hyde |

The funders had no role in study design, data collection and interpretation, or the decision to submit the work for publication.

## Author contributions

Saige Rutherford, Conceptualization, Data curation, Formal analysis, Methodology, Resources, Software, Validation, Visualization, Writing – original draft, Writing – review and editing; Charlotte Fraza, Seyed Mostafa Kia, Thomas Wolfers, Mariam Zabihi, Pierre Berthet, Methodology, Software, Writing – review and editing; Richard Dinga, Methodology; Amanda Worker, Serena Verdi, Johanna MM Bayer, Methodology, Writing – review and editing; Derek Andrews, Laura KM Han, Paola Dazzan, Phillip McGuire, Aart Schene, David Amaral, Resources, Writing – review and editing; Roel T Mocking, Chandra Sripada, Ivy F Tso, Elizabeth R Duval, Soo-Eun Chang, Brenda WJH Penninx, Mary M Heitzeg, S Alexandra Burt, Luke W Hyde, Christine Wu Nordahl, Ole A Andreasssen, Lars T Westlye, Roland Zahn, Funding acquisition, Resources, Writing – review and editing; Henricus G Ruhe, Conceptualization, Funding acquisition, Resources, Supervision, Writing – original draft, Writing – review and editing; Christian Beckmann, Conceptualization, Funding acquisition, Methodology, Project administration, Resources, Supervision, Writing – review and editing; Andre F Marquand, Conceptualization, Data curation, Formal analysis, Funding acquisition, Investigation, Methodology, Project administration, Resources, Software, Supervision, Visualization, Writing – original draft, Writing – review and editing

## Author ORCIDs

Saige Rutherford ![ORCID] http://orcid.org/0000-0003-3006-9044
Seyed Mostafa Kia ![ORCID] http://orcid.org/0000-0002-7128-814X
Pierre Berthet ![ORCID] http://orcid.org/0000-0002-6878-6842
Johanna MM Bayer ![ORCID] http://orcid.org/0000-0003-4891-6256
Chandra Sripada ![ORCID] http://orcid.org/0000-0001-9025-6453
Andre F Marquand ![ORCID] http://orcid.org/0000-0001-5903-203X

## Ethics

Human subjects: Ethical approval for the public data were provided by the relevant local research authorities for the studies contributing data. For full details see the main study publications given in

the main text. For all clinical studies, approval was obtained via the local ethical review authorities, i.e.: TOP: Regional Committee for Medical & Health Research Ethics South East Norway. Approval number: 2009/2485- C, KCL: South Manchester NHS National Research Ethics Service. Approval number: 07/H1003/194. Delta: The local ethics committee of the Academic Medical Center of the University of Amsterdam (AMC-METC) Nr.:11/050, UMich_IMPS: University of Michigan Institution Review Board HUM00088188, UMich_SZG: University of Michigan Institution Review Board HUM00080457, UMich_MLS: University of Michigan Institution Review Board HUM00040642, UMich_CWS: MSU Biomedical and Health Institutional Review Board (BIIRB) IRB#09-810, UMich_MTWiNS: University of Michigan Institution Review HUM00163965, UCDavis: University of California Davis Institutional Review Board IRB ID: 220915, 592866, 1097084.

### Decision letter and Author response
Decision letter https://doi.org/10.7554/eLife.72904.sa1
Author response https://doi.org/10.7554/eLife.72904.sa2

## Additional files

### Supplementary files
• Supplementary file 1. Full Sample Description, per site, in the train, test, and clinical validation sets.

• Supplementary file 2. Manual Quality Checked (mQC) Sample Description, per site, train and test sets.

• Supplementary file 3. Transfer Test Set Sample Descriptions, per site. Data were downloaded from https://openneuro.org/ (*Markiewicz et al., 2021*) and are shared on GitHub along with code to transfer to demonstrate how to re-use models on new data.

• Transparent reporting form

### Data availability
All pre-trained models and code for transferring to new sites are shared online via GitHub (https://github.com/predictive-clinical-neuroscience/braincharts, copy archived at swh:1:rev:ee2b7ebcb-46bab0f302f73f8d6fc913f63fccda5). We have also shared the models on Zenodo (https://zenodo.org/record/5535467#.YVRECmYzZhF).

The following dataset was generated:

| Author(s) | Year | Dataset title | Dataset URL | Database and Identifier |
|---|---|---|---|---|
| Rutherford S, Andre M | 2021 | Braincharts | https://zenodo.org/record/5535467#.YVQlVmYzZb8 | Zenodo, 10.5281/zenodo.5535467 |

The following previously published datasets were used:

| Author(s) | Year | Dataset title | Dataset URL | Database and Identifier |
|---|---|---|---|---|
| National Institute of Mental Health | 2020 | ABCD | https://nda.nih.gov/abcd/query/abcd-curated-annual-release-2.0.html | nih, 10.15154/1503209 |

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
