## [Editor Report]

This manuscript is of broad interest to the neuroimaging community. It establishes a detailed reference model of human brain development and lifespan trajectories based on a very large data set, across many cortical and subcortical brain regions. The model not only explains substantial variability on test data, it also successfully uncovers individual differences on a database of psychiatric patients that, in addition to group-level analyses, may be critical for diagnosis, thereby demonstrating high clinical potential. It presents a clear overview of the data resource, including detailed evaluation metrics, and makes code, models and documentation directly available to the community.

---

## [Decision Letter]

**Decision letter after peer review:**

Thank you for submitting your article "Charting Brain Growth and Aging at High Spatial Precision" for consideration by *eLife*. Your article has been reviewed by 3 peer reviewers, and the evaluation has been overseen by Chris Baker as the Senior and Reviewing Editor. The following individuals involved in review of your submission have agreed to reveal their identity: Bernd Taschler (Reviewer #1); Oscar Esteban (Reviewer #2); Todd Constable (Reviewer #3).

As you will see, all three reviewers are very enthusiastic about this work and have some excellent suggestions for strengthening the manuscript that will require some additional analyses.

Essential revisions:

The comments from the three reviewers are highly consistent and identify three main areas where the manuscript can be improved to increase the strength of the results and the utility of the resource.

1) All reviewers noted concerns about the current evidence for generalization of the findings. The authors should include additional cross-validation tests across sites.

2) Related to point (1), the bulk of the data come from the UK Biobank. The implications and potential limitations caused by this should be more fully discussed, although the additional cross-validation analyses will help with this.

3) The manual QC is very impressive, but the whole process could be described in more detail to enable others to reproduce such a QC.

*Reviewer #1 (Recommendations for the authors):*

This is a highly valuable resource that will hopefully grow further in the future. The manuscript is well written and data and results are presented in a clear and detailed way. I especially commend the authors on making their code easy to run, understandable and truly accessible.

One aspect that, in my opinion, would strengthen the paper is the inclusion of a more comprehensive evaluation on unseen data across sites. With clinical applications in mind where a small, in-house data set is compared to the reference models, it would be useful to understand how much variation is to be expected from scanner/site differences alone. A comparison of the existing evaluation metrics with a scenario in which the models are trained on one set of sites (or even just UKB alone) and tested on a separate set of data that does not include any of the training sites would increase the interpretability of the current results.

Several recent studies have found recruitment and selection bias in the UKB with respect to the general population and even within the imaging cohort compared to the full 500k. Although briefly mentioned in the limitations, this could be expanded further by discussing recent findings.

*Reviewer #3 (Recommendations for the authors):*

While the numbers are probably sufficient that it doesn't matter – it seems that the train and test sets were only split once – and then the results presented. Proper form might be to randomly split the train/test set multiple times to obtain distributions. It would be much stronger statistically if this was repeated. If this was already repeated then it should be made clearer. The wording refers to train and test set(s) with sets being plural, but I could not find anything explicitly stating how many times this was repeated.

The data shown in Figure 1 might be better served by splitting this into multiple figures. In A it is impossible to read the y-axis. In C and D the caption states that the lines are centiles of variation but it doesn't say what centiles (for example do they match the centiles of pediatric growth charts 0.4th, 2, 9th, 25, 50th etc?) – this should be stated.

Figure 1C shows whole cortex results, while D shows subcortical. It would be nice to show data for some cortical brain regions – or even summarized for lobes instead of just whole brain.

For regions, it would be reassuring to see that the development curves for PFC for example, agree with the previous literature. Or even show that different regions have different temporal growth charts. Similarly, the work could be put in context with the work of Toga et al., Trends Neurosci, 2006 – mapping brain maturation. Or the work of Pigoni et al., Eur Neuropsychopharm, 2021 where they show (in a large sample) that cortical thickness changes in the temporal lobes can be used in classification of first episode psychosis. While the authors state that a thorough analysis of these curves is beyond the scope (and I agree) it would be helpful to have some text that confirms these curves (for healthy or diseased brains) agree with past literature.

Overall I am enthusiastic to see this work published.

---

## [Author Response]

Essential revisions:The comments from the three reviewers are highly consistent and identify three main areas where the manuscript can be improved to increase the strength of the results and the utility of the resource.1) All reviewers noted concerns about the current evidence for generalization of the findings. The authors should include additional cross-validation tests across sites.

We thank the reviewers for this suggestion and have addressed the concern regarding generalizability in several ways. First, we ran an additional 10 randomized train/test splits of the data in the full sample. These new analyses show the stability of our models, as there is very little variation in the evaluation metrics across all 10 splits. These results are visualized in Figure 3—figure supplement 2. However, the static Figure 3—figure supplement 2 is challenging to read, simply because there are many brain regions fit into a single plot. Therefore, we also created an online interactive visualization tool (https://brainviz-app.herokuapp.com/) that shows the image of the brain region and the explained variance when you hover over a point (see Author response image 1 and Video 1). This interactive visualization was created for all supplemental tables for easier exploration and interpretations and we now recommend this tool as the primary method to interrogate our findings interactively.

**Author response image 1. sa2fig1:** Example of the online interactive visualizations created to help interpret the evaluation metrics. This interactive figure was created for each evaluation metric (EV, MSLL, skew, and kurtosis) and all test sets (full controls, mQC controls, clinical, and transfer).

Second, we updated and expanded the transfer data set to include 6 open datasets from OpenNeuro.org (N=546) and we provide this example dataset on our GitHub with the transfer code (https://colab.research.google.com/github/predictive-clinical-neuroscience/braincharts/blob/master/scripts/apply_normative_models.ipynb). This simultaneously provides a more comprehensive evaluation of the performance of our model on unseen data and more comprehensive walk-through for new users applying our models to new data (sites unseen in training).

Finally, we added per-site evaluation metrics (Figure 3—figure supplement 3) to demonstrate that performance is relatively stable across sites and not driven by a single large site (i.e., UKB). As site is strongly correlated with age, these visualizations can also be used to approximate model performance at different age ranges (i.e., 9–10-year-old performance can be assessed by looking at ABCD sites evaluation metrics, and 50–80-year-old performance can be assessed by looking at UKB evaluation metrics). Moreover, we would also like to emphasize that we should not expect that all sites achieve the same performance because the sampling of the different sites is highly heterogeneous in that some sites cover a broad age range (e.g., OASIS, UKB) whereas other sites have an extremely narrow age range (e.g., ABCD).

2) Related to point (1), the bulk of the data come from the UK Biobank. The implications and potential limitations caused by this should be more fully discussed, although the additional cross-validation analyses will help with this.

The responses to revision 1 should help to address this, in that we show per site evaluation metrics, cross validation, and additional transfer examples. These additional analyses show that the model performance is not driven solely by the UKB sample. However, we agree with this comment and have also updated the limitation section regarding the overrepresentation of UKB and included a statement regarding the known sampling bias of UKB.

“We also identify limitations of this work. We view the word “normative” as problematic. […] By sampling both healthy population samples and case-control studies, we achieve a reasonable estimate of variation across individuals, however, downstream analyses should consider the nature of the reference cohort and whether it is appropriate for the target sample.”

3) The manual QC is very impressive, but the whole process could be described in more detail to enable others to reproduce such a QC.

We have added further details regarding the quality checking procedure to the methods section (following reviewer 2 comment 3a-c suggestions) and improved the clarity of directions for implementing the scripts, including an interactive link to view an example of the manual QC environment, on the QC GitHub page to enable others to reproduce our manual QC pipeline.

“Quality control (QC) is an important consideration for large samples and is an active research area (Alfaro-Almagro et al., 2018; Klapwijk et al., 2019; Rosen et al., 2018). […] We separated the evaluation metrics into full test set (relying on automated QC) and mQC test set in order to compare model performance between the two QC approaches and were pleased to notice that the evaluation metrics were nearly identical across the two methods.”

Reviewer #1 (Recommendations for the authors):This is a highly valuable resource that will hopefully grow further in the future. The manuscript is well written and data and results are presented in a clear and detailed way. I especially commend the authors on making their code easy to run, understandable and truly accessible.One aspect that, in my opinion, would strengthen the paper is the inclusion of a more comprehensive evaluation on unseen data across sites. With clinical applications in mind where a small, in-house data set is compared to the reference models, it would be useful to understand how much variation is to be expected from scanner/site differences alone. A comparison of the existing evaluation metrics with a scenario in which the models are trained on one set of sites (or even just UKB alone) and tested on a separate set of data that does not include any of the training sites would increase the interpretability of the current results.

As noted, we have addressed this concern in response to Essential revisions 1) and 2) above.

Several recent studies have found recruitment and selection bias in the UKB with respect to the general population and even within the imaging cohort compared to the full 500k. Although briefly mentioned in the limitations, this could be expanded further by discussing recent findings.

We have addressed this concern in response to Essential revision 2) above.

Reviewer #3 (Recommendations for the authors):While the numbers are probably sufficient that it doesn't matter – it seems that the train and test sets were only split once – and then the results presented. Proper form might be to randomly split the train/test set multiple times to obtain distributions. It would be much stronger statistically if this was repeated. If this was already repeated then it should be made clearer. The wording refers to train and test set(s) with sets being plural, but I could not find anything explicitly stating how many times this was repeated.

We have addressed this comment under Essential revision 1) above. Regarding nomenclature, in our previous version the plural use of test sets previously referred to the full test set, mQC test set, transfer test set, and clinical test set (Table 1). However, we thank the reviewer for pointing out that this is open to misinterpretation and have included specific test set names when referring to them instead of just using “test sets”. We also have included a resampling of the full controls test set to address the generalizability concern. Additional details on this analysis are in response to Essential revision 1 above.

The data shown in Figure 1 might be better served by splitting this into multiple figures. In A it is impossible to read the y-axis. In C and D the caption states that the lines are centiles of variation but it doesn't say what centiles (for example do they match the centiles of pediatric growth charts 0.4th, 2, 9th, 25, 50th etc?) – this should be stated.

We have increased the resolution and font size in Figure 1 and added labels to the centiles.

Figure 1C shows whole cortex results, while D shows subcortical. It would be nice to show data for some cortical brain regions – or even summarized for lobes instead of just whole brain.For regions, it would be reassuring to see that the development curves for PFC for example, agree with the previous literature. Or even show that different regions have different temporal growth charts. Similarly, the work could be put in context with the work of Toga et al., Trends Neurosci, 2006 – mapping brain maturation. Or the work of Pigoni et al., Eur Neuropsychopharm, 2021 where they show (in a large sample) that cortical thickness changes in the temporal lobes can be used in classification of first episode psychosis. While the authors state that a thorough analysis of these curves is beyond the scope (and I agree) it would be helpful to have some text that confirms these curves (for healthy or diseased brains) agree with past literature.

We have included a frontal cortex ROI trajectory in Figure 1 D and there is another cortical ROI shown (calcarine). Also on our studies GitHub(https://github.com/predictive-clinical-neuroscience/braincharts/tree/master/metrics), we now include images of the trajectory for every ROI so that users can further explore these results if they wish to see a certain ROI that is not represented in Figure 1 as there is not enough space to show all brain regions. We have included additional citations to prior work mapping trajectories in the Results section.